# PDX Models: A Versatile Tool for Studying the Role of Myeloid-Derived Suppressor Cells in Breast Cancer

**DOI:** 10.3390/cancers14246153

**Published:** 2022-12-13

**Authors:** Morten F. Gjerstorff, Sofie Traynor, Odd L. Gammelgaard, Simone Johansen, Christina B. Pedersen, Henrik J. Ditzel, Mikkel G. Terp

**Affiliations:** 1Department of Cancer and Inflammation Research, Institute for Molecular Medicine, University of Southern Denmark, J. B. Winsløws Vej 25, 3, DK-5000 Odense, Denmark; 2Department of Oncology, Odense University Hospital, DK-5000 Odense, Denmark; 3Academy of Geriatric Cancer Research (AgeCare), Odense University Hospital, DK-5000 Odense, Denmark

**Keywords:** PDX models, MDSC, breast cancer

## Abstract

**Simple Summary:**

Myeloid-derived suppressive cells (MDSCs) are important for the progression of human tumors and represent potential targets for novel therapeutic strategies in breast and other cancers. To develop such strategies, pre-clinical models mimicking patient tumors are needed. In this study, we demonstrate that tumor models established by transplanting breast cancer patient tumor biopsies into the mammary tissues of mice represent excellent tools for studying and targeting MDSCs. Using molecular and genetic analyses, we show that these patient-derived xenograft (PDX) tumors produce signaling proteins that actively recruit MDSCs to the tumors, with differences between models that reflect those seen in breast cancer patients. Furthermore, MDSCs were associated with the spread of cancer cells to distant sites in the mice, similar to what is observed in patients. In conclusion, the use of breast cancer PDX models may greatly facilitate the development of novel therapeutic strategies for breast cancer and other cancer types.

**Abstract:**

The pivotal role of myeloid-derived suppressive cells (MDSCs) in cancer has become increasingly apparent over the past few years. However, to fully understand how MDSCs can promote human tumor progression and to develop strategies to target this cell type, relevant models that closely resemble the clinical complexity of human tumors are needed. Here, we show that mouse MDSCs of both the monocytic (M-MDCS) and the granulocytic (PMN-MDSC) lineages are recruited to human breast cancer patient-derived xenograft (PDX) tumors in mice. Transcriptomic analysis of FACS-sorted MDSC-subpopulations from the PDX tumors demonstrated the expression of several MDSC genes associated with both their mobilization and immunosuppressive function, including *S100A8/9*, *Ptgs2*, *Stat3*, and *Cxcr2*, confirming the functional identity of these cells. By combining FACS analysis, RNA sequencing, and immune florescence, we show that the extent and type of MDSC infiltration depend on PDX model intrinsic factors such as the expression of chemokines involved in mobilizing and recruiting tumor-promoting MDSCs. Interestingly, MDSCs have been shown to play a prominent role in breast cancer metastasis, and in this context, we demonstrate increased recruitment of MDSCs in spontaneous PDX lung metastases compared to the corresponding primary PDX tumors. We also demonstrate that T cell-induced inflammation enhances the recruitment of MDSC in experimental breast cancer metastases. In conclusion, breast cancer PDX models represent a versatile tool for studying molecular mechanisms that drive myeloid cell recruitment to primary and metastatic tumors and facilitate the development of innovative therapeutic strategies targeting these cells.

## 1. Introduction

Myeloid-derived suppressor cells (MDSC) are a cell type that exhibit potent immunosuppressive activity [1]. According to their origin from the granulocytic or monocytic myeloid lineages, MDSCs have been divided into two major groups in humans and mice, granulocytic/polymorphonuclear MDSCs (PMN-MDSC), and monocytic MDSCs (M-MDSCs), respectively. MDSCs are implicated in regulating immune responses in a wide range of pathological conditions, including chronic infection, autoimmunity, and cancer. Activation and differentiation of the MDSCs in these pathological conditions arises from prolonged stimulation by myeloid growth factors and inflammatory mediators [1]. Well-described signals associated with MDSC recruitment and activation include granulocyte-macrophage colony stimulating factor (GM-CSF, also known as CSF2), granulocyte colony stimulating factor (G-CSF, also known as CSF3), interleukin (IL)-6, IL-1β, and C-X-C motif chemokine (CXCL)-1, -5, and -8 [2,3]. In mice, the standard phenotypic markers for PMN-MDSC are CD11b^+^Ly6G^+^Ly6C^−^, while M-MDSC is defined as CD11b^+^Ly6G^−^Ly6C^+^ [1]. Inconveniently, no phenotypic cell surface markers can distinguish neutrophils from PMN-MDSC and monocytes from M-MDSC, and the main characteristic that defines MDSC is their ability to suppress other immune cells, including T and B cells and natural killer (NK) cells. M-MDSC and PMN-MDSCs share immunosuppressive properties such as upregulation of signal transducers and activators of transcription (STAT) 3, expression of arginase (arg) 1, S100A8/9, prostaglandin E_2_ (PGE2), IL10 and transforming growth factor (TGF)-β, and production of reactive oxygen species (ROS) and nitric oxide (NO) [4,5]. The mechanisms of action of these factors on T cell function in the tumor microenvironment (TME) are complex and diverse. ROS and IL-10 have been shown to restrict T cell activation and proliferation [6,7], whereas TGF-β directly targets cytotoxic T cell functionality by reducing the expression of perforin, granzyme A, granzyme B, Fas ligand, and interferon γ [8]. In the context of cancer, MDSCs have recently been identified as major contributors to tumor progression and metastasis development [9,10,11] and, in most studies, are associated with a poor prognosis and a poor response to therapy, including chemotherapy, radiation, and immunotherapy, across multiple tumor types [12,13,14]. Several reports have investigated the potential of reducing MDSC tumor infiltration to increase T cell abundance and sensitize tumors to anti-PD-1 and CAR-T cell immunotherapy [15,16,17,18]. For instance, in breast cancer, it was shown that MDSC-enriched tumors display lower T cell activity and resistance to immune checkpoint blockade [19,20,21]. Hence, the inverse relationship between MDSC infiltration and anti-tumor T cell responses suggests a direct MDSC-mediated suppression of cytotoxic T cells in tumors. Here, we demonstrate that human patient-derived xenograft (PDX) models provide a robust in vivo model system to study molecular mechanisms that drive MDSC recruitment to tumors and to develop and evaluate novel therapeutic strategies to target this cell type to enhance the effect of anti-cancer therapies, including immunotherapies.

## 2. Materials and Methods

### 2.1. In Vivo Models

To generate breast cancer PDX models, we used tissue biopsies from TNBC patients undergoing routine treatment at Odense University Hospital or frozen stocks of PDX tumors implanted with Matrigel (Sigma-Aldrich, Søborg, Denmark) into the mammary fat pad of 7–8-week-old female NOG CIEA mice (NOD.Cg-*Prkdc^scid^ Il2rg^tm1Sug^*/JicTac, Taconic, Ry, Denmark). PDX tumor samples were stored in DMEM with 10% DMSO in liquid nitrogen and used in low passage.

For the adoptive T cell transfer model, we initially injected MDA-MB-231 cells intravenously (i.v.) to establish pulmonary metastases. After 4 weeks, partially HLA-matched human T cells were injected i.v. and treatment with intraperitoneal injection of 200 μg anti-PD1 antibodies twice a week was initiated and continued for 3 weeks.

All animal experiments were approved by the Experimental Animal Committee of the Danish Veterinary and Food Administration and performed at the animal core facility at the University of Southern Denmark. Mice were acclimatized for at least one week before experiments were initiated. The mice were housed under specific pathogen-free conditions with ad libitum food and water. All experimental groups were housed under similar conditions. Due to the limited availability of tissues and the complexity of the PDX models, the number of animals in several of the experiments was limited.

### 2.2. Immunohistochemistry

Formalin-fixed, paraffin-embedded, tumor sections (4 μm) were cut with a microtome, mounted on ChemMateTM capillary gap slides (Dako, Glostrup, Denmark), dried at 60 °C, deparaffinized, and hydrated. Hydrogen peroxide (1.5%) in TBS buffer (pH 7.4) was used for blocking endogenous peroxidase, and antigen retrieval was achieved by microwave boiling in TEG buffer (Dako) for 15 min. Sections were incubated with the primary antibody: anti-mouse CD11b (ab133357, Abcam); anti-human cytokeratin (M3515, Dako, Clone AE1/AE3); and anti-human CD8 (108M, Sigma-Aldrich) at room temperature for 32 min before detection with the OptiView DAB IHC detection kit (760-700, Vienna Medical Systems).

### 2.3. Purification of Neutrophils and MDSCs

Neutrophils from naïve non-tumor-bearing mice were harvested from six- to eight-week-old female NOG mice. The bone marrow of the femurs and tibias was isolated in sterile Hank’s buffered salt solution (HBSS 1X) (Gibco). Bone marrow cells were flushed in HBSS, and after centrifugation (1500 rpm, 5 min, 4 °C), the cells were resuspended for 3 min on ice in an ammonium-chloride-potassium (ACK) buffer (Sigma-Aldrich) and centrifuged (1500 rpm, 5 min, 4 °C) before being washed twice with HBSS 1X (1500 rpm centrifugation was used for the washing steps, 5 min, 4 °C). MDSC were isolated from splenocytes from tumor-bearing NOG mice.

Neutrophils/MDSC were then separated from mononuclear cells by plating 2 mL of the cell suspension onto a Percoll gradient (GE Healthcare, Brøndby, Denmark) consisting of 3 mL of 81% Percoll under 3 mL of 62% Percoll, followed by centrifugation at 2500 rpm for 20 min at 4 °C. The middle layer containing the neutrophils was washed twice in HBSS 1X, and cells were resuspended in serum-free DMEM (Gibco).

### 2.4. Human Interferon Gamma (IFNγ) Measurement

IFNγ secretion from T cells cultured with CD3/CD28 beads for 48 h was measured using the Human IFNγ uncoated ELISA kit (ThermoFisher Scientific, Roskilde, Denmark, 88-7316-88) according to the manufacturer’s instructions. The plate was read using a SpectraMax i3x platform (Molecular Devices) and the difference in wavelength (450–570 nm) was calculated.

### 2.5. Immunofluorescence

Formalin-fixed, paraffin-embedded, tumor sections (8 μm) were cut with a microtome, mounted on ChemMateTM capillary gap slides (Dako), dried at 60 °C, deparaffinized and hydrated. Hydrogen peroxide (1.5%) in TBS buffer (pH 7.4) was used for blocking endogenous peroxidase, and antigen retrieval was achieved by microwave boiling in TEG buffer (Dako) for 15 min. Sections were blocked with an Fc receptor blocker (BD Bioscience, #553141) and incubated with 1X blocking buffer (2.5% BSA, 0.1% Triton X-100 in PBS). The sections were incubated overnight at 4 °C with anti-myeloperoxidase (R&D System, AF3667, 1:100) in 0.5X blocking buffer. After three washes with PBS, the sections were incubated with Alexaflour488-conjugated secondary antibodies (Invitrogen; A10034 1:150) in 0.5x blocking buffer for 45 min in the dark at room temperature. After two washes with PBS and one with water, sections were counterstained with DAPI (#D1306, Thermo Fisher Scientific) and rinsed in water, and the slides were mounted onto coverslips using mounting media (Invitrogen, #P36930).

### 2.6. Tumor Dissociation

Tumors were harvested for dissociation into single-cell suspensions, as previously described [22]. In brief, minced tumors were incubated for 45 min in dissociation buffer [2 mg/mL collagenase IV (Merck Life Science, Søborg, Denmark, #C5138)] supplemented with 4 U/mL DNAse I (Thermo Scientific, #EN0521) with constant low agitation at 37 °C and subsequently pipetted up and down for 2 min using a 5 mL pipette. Following a 1x wash in complete DPBS (DPBS with 5% FBS), the cell suspension was filtered using a 70 μm cell strainer (BD, # 352350), and the cells were resuspended in 2 mL of red blood cell lysing buffer (155 mm NH_4_Cl, 12 mm NaCO_3_, and 0.1 mm EDTA) and gently mixed for 1 min at room temperature. Following 2x wash in complete DPBS, the single cells were counted and stained for flow cytometry.

### 2.7. Flow Cytometry

Single cell suspensions of PDX tumor cells were stained with Brilliant Violet 605 anti-mouse CD45 (Biolegend #103139); BB515/FITC anti-mouse CD11b (BD Bioscience #564454); MHCII-APC (MBL International Corporation FP20868002); PE anti-mouse Ly6C (BD Bioscience #560592); and BV421 anti-mouse Ly6G (BD Bioscience #562737). Cells were counterstained with the live/dead fixable near-IR cell death marker (Thermo Fischer Scientific). The cells were analyzed on a BD *FACSAria*™ III (BD bioscience). Data was analyzed using FlowJo software (version 10, FlowJo).

For cell sorting experiments, single cell suspensions were stained as described above, and the two populations (CD45^+^CD11b^+^MHCII^−^LY6C^−^Ly6G^−^) and (CD45^+^CD11b^+^MHCII^−^LY6C^−^Ly6G^+^) were sorted in TRI reagent (Sigma-Aldrich) using a BD *FACSAria*™ III (BD bioscience).

### 2.8. RNA Sequencing

RNA was purified using RiboZol (VWR) or TRI Reagent (Sigma-Aldrich). For tissues, this step included homogenization using 2.8 mm zirconium oxide beads (Precellus) and a Precellus 24 homogenizer (3 × 15 s, 6500 rpm). Purified RNA was prepared for sequencing on the Illumina NovaSeq 6000 Sequencing Platform using the NEBNext Poly(A) mRNA Magnetic Isolation Module (New England Biolabs, Herlev, Denmark, E7490L) and the NEBNext Ultra II DNA Library Prep Kit for Illumina (New England Biolabs, E7645L) with unique dual indexes according to the manufacturer’s instructions. The quality of raw sequencing reads was assessed using FASTQC (Babraham Bioinformatics, Braham Institute, Cambridge, Great Britain), and adaptor sequences were removed using the FASTX toolkit. Trimmed and filtered sequencing reads were aligned to the human (hg38) and mouse (mm10) genomes using Spliced Transcripts Alignment to a Reference (STAR) software with default parameters [23]. Tags in exons were counted using iRNA-seq [24], and differential expression between the P5 and P6 populations was determined using DESeq2 [25]. Differentially expressed genes were defined as those having FDR ≤ 0.05 and a log2 fold change > 1.0 in either direction. 

## 3. Results

### 3.1. Establishment of PDX Models for MDSC Studies

The purpose of this study was to characterize the mobilization and recruitment of MDSCs in breast cancer PDX models. All PDX tumors were established from patients with triple-negative breast cancer (TNBC) and were generated using biopsies from either treatment-naïve primary tumors or metastases treated with multiple lines of chemotherapy (Figure 1a). We observed little variance of tumor growth between the different tumors of the same PDX model, while tumor growth between the different PDX models was very heterogeneous (Figure 1b). We found no correlation between tumor growth rate and the origin of the PDX model or lines of therapy prior to the establishment of the PDX models. Hematoxylin and eosin staining showed that all the tumors were well-defined and solid, with distinct pleomorphic nuclei and large nucleoli (Figure 1c). Furthermore, while areas with necrosis and fibrosis were generally observed in all the PDX tumors, these phenotypic characteristics were less pronounced in the PDX9228 compared to the other models. Thus, all the PDX tumors were histologically similar and reflected clinical tumors.

### 3.2. Characterization of PDX Models Reveals Heterogenic Infiltration of Myeloid Cells

To elucidate the myeloid immune cell compartment in PDX tumors, we first evaluated the infiltration of myeloid cells in the different PDX tumors using flow cytometry and a panel of immune markers that allowed phenotypic evaluation of the infiltrated myeloid cell populations in the resected tumors. The PDX tumor size and timepoint upon harvest are shown in Appendix A. We manually analyzed the flow cytometry data using an exclusion-based gating strategy (Figure 2a). The level of immune infiltration (CD45^+^ cells) in tumors differed among the PDX models (PDX5474: ~60%, PDX4582: ~25%, PDX9228: ~35%, PDX1004: ~65%, and PDX1002: ~75%) but was dominated by CD11b^hi^ myeloid cells (91.5% ± 6.7%, Figure 2a). Among the CD45^+^ cells, we observed no MHCII-positive cells, indicating that no mature macrophages or dendritic cells were present in the PDX tumors. While all PDX tumors contained both M-MDSC (CD11b^+^MHCII^−^Ly6C^+^Ly6G^−^) and PMN-MDSC (CD11b^+^MHCII^-^Ly6C^−^Ly6G^+^), there seemed to be no consistent correlation between the presence of the two populations in the PDX models (Figure 2b). For instance, while PDX5474 had the highest infiltration of PMN-MDSC (71.7 ± 1.63%, relative to CD11b^+^) (Figure 2c), this PDX model had the lowest infiltration of M-MDSC (4.72 ± 2%, relative to CD11b^+^) (Figure 2c). In addition, while PDX1002 and PDX1004 had a very low infiltration of PMN-MDSC compared to the other PDX models (Figure 2c), the infiltration of M-MDSCs in these two PDX models was similar to the other PDX models (Figure 2c). This strongly suggests that in the PDX model, intrinsic factors regulate the recruitment of the different subtypes of MDSCs. 

Next, we further analyzed the infiltration of PMN-MDSCs myeloid cells using immunofluorescence staining of MPO, a marker associated with neutrophils and PMN-MDSCs. The number of MPO-positive PMN-MDSCs was quantified across multiple individual tumors and demonstrated a very similar infiltration of PMN-MDSCs in the different PDX tumors as was observed in the FACS experiments (Figure 2d). The PDX tumor size and timepoint upon harvest are shown in Appendix A. Finally, we also investigated the spatial distribution of PMN-MDSC in the PDX models. In PDX9228, PDX4582, and PDX1002, most PMN-MDSCs were located at the tumor border with a limited and diffuse infiltration in the tumor core (Figure 2e). In contrast, while many myeloid cells were also located at the tumor border in PDX5474, this PDX model also presented with high core infiltration (Figure 2e). In the PDX1004, the limited infiltration of PMN-MDSC seemed to be more homogeneous throughout the tumors. Thus, the variation in the spatial distribution of the myeloid cells in the different PDX tumors also supported PDX-intrinsic recruitment of the myeloid cells.

### 3.3. MDSC Infiltration Correlates with the Expression of Myeloid Cell-Recruiting Chemokines

Next, we wanted to investigate the mobilization and recruitment of MDSCs in more detail using bulk RNA sequencing of the PDX tumors. We first confirmed the pattern of MDSC infiltration among the three PDX models PDX5474, PDX4582, and PDX9228 by analyzing the expression of a panel of 11 previously described MDSC associated genes in mice involved in recruitment (*Cxcr2* and *Csf3r*), activation (*IL1b* and *Ifitm*), and immunosuppression (*S100A8/9* and *Cd84*) [26]. Consistent with the high infiltration of PMN-MDSC in PDX5474 (Figure 2), we found that this model had a relatively high expression of all genes within the panel compared to PDX4582 and PDX9228 (Figure 3a). 

MDSCs are known to be actively mobilized and recruited to tumors following the expression of various chemokines by the cancer cells [27]. We aimed at elucidating factors responsible for the observed PDX model-induced intrinsic recruitment of MDSCs. Intriguingly, the PDX5474 expressed the highest level of several chemokines, including CXCL1, IL-1β, and CSF2, all well-known for their involvement in MDSC recruitment and activation [27,28,29,30] (Figure 3b). These data verify that MDSCs are actively recruited to PDX tumors and that the recruitment and infiltration of the MDSCs are dependent on a tumor-specific expression of chemokines and cytokines. To explore the expression of these cytokines in human patient tumors, we analyzed a cohort of breast cancer patients with the basal subtype, which significantly overlaps with TNBC. While this analysis revealed a broad expression pattern of CXCL1, CSF2, and IL-1β, we observed that 27%, 23%, and 11% of the patients, respectively, exhibited high expression, defined as exhibiting expression levels three times above the median expression of the cohort (Appendix A). Thus, similar to our observations in the PDX models, it seems that while most patients exhibit a low to medium expression of CXCL1, CSF2, and IL-1β, a fraction of breast cancer patients exhibit high expression levels of these cytokines.

### 3.4. Tumor-Associated PMN-MDSC in Breast Cancer PDX Models Display MDSC Expression Profiles

To characterize the PMN-MDSCs in the PDX models in more detail and investigate the main molecular features previously shown to regulate their immunosuppressive functions, we performed transcriptional profiling of PMN-MDSCs (Ly6C^−^LyG6^+^ (P5), Figure 4a) and non-MDSC (Ly6C^−^LyG6^−^ (P6), Figure 4a) cells purified using FACS sorting for three PDX models (PDX9228, PDX1004, and PDX1002) (Figure 4a). As expected, transcriptional profiling clearly separated the PMN-MDSC from the non-MDSC in all three PDX models (Figure 4b). Comparing the three PDX models further revealed a large overlap of the genes that were significantly more highly expressed in the PMN-MDSC vs. the non-MDSC (Figure 4c), indicating that the PMN-MDSCs in the different PDX models likely represent a distinct cell population with shared biochemical features. The non-MDSC populations were characterized by high expression of genes including *Irf8*, *Cx3cr1*, and *Csf1* that are associated with monocyte differentiation (Figure 4b). In agreement, CD11b^+^ Ly6C^−^ cells have been proposed to represent a monocyte-dendritic cell progenitor-like (MDPs) population with a high expression of *Cx3cr1* [31]. Furthermore, Irf8 regulates myeloid lineage diversification, and the expression of Irf8 promotes the differentiation of monocytes and dendritic cells. Collectively, this suggests that the non-MDSC population is at least partly composed of immune cells from the monocytic lineage and, importantly, represents a population that is clearly different from the MDSC immune cell type. 

As mentioned above, MDSC are mainly defined by their ability to inhibit immune responses, including those mediated by T cells, B cells, and natural killer (NK) cells. Interestingly, the transcriptomic analysis of the PMN-MDSC population revealed a very high expression of many genes, supporting their suppressive phenotype (Figure 4d). These included the genes for *S100a8/9*, *Arg2*, and prostaglandin-endoperoxide synthase 2 (*Ptgs2*) (Figure 4d). We also identified several genes supporting MDSC recruitment that were highly expressed in this population, including C-X-C motif chemokine receptor 2 (*Cxcr2)* and *Csf3*, as well as increased levels of the transcription factors *Stat3* and CCAAT enhancer binding protein beta (*Cebpb)* which are involved in regulating the expansion of MCSC in cancer and the expression of immunosuppressive factors [1,32] (Figure 4d). Finally, the PMN-MDSC population also exhibited high expression of genes encoding chemokines associated with inflammation, including *csf1*, *Il1β*, *Cxcl2*, and *Cxcl3* (Figure 4d). Collectively, this transcriptomic analysis demonstrates that the CD11b^+^Ly6C^−^LyG6^+^ cell population recruited to PDX models is, in fact, PNN-MDSCs with an immunosuppressive phenotype.

### 3.5. Recruitment of MDSCs to Spontaneous PDX Lung Metastases 

MDSCs have been shown to play an important role in breast cancer metastasis, and thus we investigated the infiltration of MDSCs in breast cancer lung metastases from PDX mouse tumors using the PDX9228 model that spontaneously metastasizes to lymph nodes and the lung. We observed that the infiltration of CD11b-positive cells in the lymph node metastases was slightly higher compared to the primary tumors, while the highest levels of CD11b-positive cells were observed in the lung metastases (Figure 5a). We further evaluated the infiltration of the CD11b-positive cells in the lung and lymph node metastases using immunofluorescent staining of the neutrophil marker, *MPO*. Many cells in the metastases were MPO-positive (Figure 5b), and when compared within the same metastatic foci, they were present at similar frequencies as CD11b-positive cells in multiple metastases (Figure 5c), indicating that the majority of the infiltrated myeloid cells are derived from a neutrophil lineage and thus likely represent PMN-MDSCs. The number of MPO-positive cells in the metastases was considerably higher compared to both the lymph node metastasis and the primary tumor (Figure 5d), which reflects the pattern of CD11b infiltration in Figure 5a. Taken together, these data strongly support the use of PDX models in studies of PMN-MDSC function in spontaneous metastasis development.

Recently, it was shown that CD8+ T cell activation following PD-1 treatment induced inflammation via the pyrin domain-containing protein 3 (NLRP3) inflammasome signaling cascade, which led to the recruitment of granulocytic myeloid-derived suppressor cells (PMN-MDSCs) that counteracted the T cell-mediated toxicity [33]. PMN-MDSCs are receiving increasing attention due to their ability to limit T cell infiltration and functionality [10,33]. To assess whether T cell-induced inflammation increased the recruitment of MDSC in a human breast cancer model system, which might provide an instrument to evaluate MDSC targeting strategies to enhance T cell-based therapies, we utilized an adoptive cell transfer (ACT) model suitable for studying anti-cancer T cell responses and PD-L1/PD-1 cancer immunotherapies [34]. We established metastasis in the lungs of mice by intravenous injection of MDA-MB-231 TNBC cells. After five weeks, partly HLA-matched human T cells were administered intravenously, and anti-PD-L1 therapy was initiated. After another three weeks, animals were sacrificed, and the lungs were examined by immunohistochemistry. In the metastases from mice treated with combined T cell and anti-PD-L1 therapy, the density of pan-cytokeratin-positive tumor cells was significantly lower compared to metastases from mice that received only T cells and control mice, indicating that the T cells were eliciting a partial response (Figure 5e,f). Interestingly, we observed highly increased levels of myeloid cells in the metastases of the combined T cell- and anti-PD-L1-treated mice compared to control mice and mice treated with T cells alone (Figure 5e). This indicates that T cell-induced inflammation enhances the recruitment of myeloid cells to metastases and demonstrates the utility of this model to investigate MDSC-targeting strategies to enhance T cell-based immunotherapies.

Finally, we have investigated the cross-reactivity and function of murine PMN-MDSC isolated from tumor-bearing mice to inhibit the activation of human T cells activated using CD3/CD28 beads. The inhibitory effect of PMN-MDSC was compared to that of naïve neutrophils isolated from non-tumor-bearing mouse bone marrow. We demonstrate that PMN-MDSC significantly inhibit the activation of T cells, as measured by IFN-gamma release, compared to T cells alone or T cells and naïve neutrophils (Figure 5g). This suggests that human cancer cells are able to mobilize murine MDSC that can inhibit the activation of T cells.

## 4. Discussion

PDX models have recently attracted substantial attention in translational cancer research, and considerable efforts to develop and characterize PDX models for cancer research are ongoing. Recent studies across different cancer types, including breast cancer, have shown that histologic, genetic, and biologic characteristics are highly comparable between PDX models and their donor tumors [35]. Furthermore, these models have been shown to be very predictive of clinical drug responses and are currently proving successful in preclinical drug evaluation and screening, biomarker identification, and personalized medicine strategies. However, only a few studies have addressed the tumor immune microenvironment in PDX models, and since the predictive value of PDX models has already been proven to be very high, these models may represent excellent potential for studies targeting the TME in breast cancer. Understanding the molecular mechanisms that cancer cells utilize to modulate the tumor microenvironment for suppression of an anti-tumor immune response will be critical to improving current approaches for cancer immunotherapy, such as checkpoint inhibition and adoptive T cell transfer, including CAR-T cell treatments. MDSCs represent one of the cell types in the TME that commonly expand in cancer patients and promote advanced tumor progression and T cell suppression in breast cancer [5]. To investigate the biological behavior of MDSCs in breast cancer and identify and develop novel strategies to target these cell populations, it is critical to have experimental models that mimic clinical tumors with respect to fundamental MDSC-associated biological processes, such as differentiation, cancer-mediated recruitment, and spatial distribution in tumors.

Here, we demonstrate that MDSCs are recruited to human breast cancer PDX models in varying degrees using both FACS analysis, immunostaining, and RNA sequencing on multiple and independent PDX tumors. All three methods supported a specific pattern of infiltration between the different PDX models and thus provided a robust investigation of the myeloid cell infiltration in PDX models. This differential infiltration of MDSCs reflects observations in clinical breast cancer tumors [36,37] and thus supports the utility of breast cancer PDX models for pre-clinical studies of MDSC behavior. Tumors secrete many different soluble factors that guide MDSC movement from the bone marrow to the tumor, including mobilizing MDSCs into the blood and recruiting MDSCs from the blood to the tumors. Interestingly, we show that the infiltration of MDSCs was correlated to the expression of various well-known MDSC-recruiting cytokines. For instance, PDX5474 tumors consistently exhibited the highest infiltration of MDSCs and concurrently the highest expression of numerous cytokines important for MDSC recruitment, including *CXCL1, CXCL8*, *CSF2* and 3, and IL-1β. PDX9228 and PDX4582 tumors exhibited high expression of *CCL5* and *CXCL5*, respectively, two cytokines shown to mobilize MDSCs from the bone marrow and infiltrate at the tumor site [38,39]. Thus, we have demonstrated that the PDX tumors express different well-known MDSC-recruiting chemokines and are able to induce inflammatory processes that drive MDSC accumulation in the tumor. Therefore, it seems that breast cancer PDX tumors support the recruitment and differentiation of MDSCs despite potential structural and functional cross-species differences in chemokine/chemokine receptor pairs. This further adds to the relevancy of PDX models for studying MDSC biology.

The MDSC cell population arises from hematopoietic myeloid progenitors in the bone marrow through a myelopoietic development program. This involves GM-CSF, G-CSF, and macrophage colony-stimulating factor (M-CSF) that dictates expansion of granulo-monocytic precursors (GMPs) via STAT3 activation [2], as well as CCAAT-enhancer-binding protein α (C/EBPα), C/EBPβ, and interferon regulatory factor 8 (IRF8) [40]. Furthermore, activation of immature cells is mainly driven by the nuclear factor-κB (NF-κB) pathway involving tumor necrosis factor-α (TNF-α), multiple interleukins, prostaglandin E2 (PGE2), S100A8/A9, and high mobility group box 1 (HMGB1) [1,41]. Many of these factors were also shown to be highly expressed in the FACS-sorted MDSC populations in this study (Figure 4a), supporting the biological relevance of immunodeficient mice as hosts for PDX tumors. The immunodeficient model used in this study lacks mature T, B, and NK cells, displays reduced complement activity, and has dysfunctional macrophages and dendritic cells, as also supported by our own FACS results showing no MHCII-positive cells in the tumors. However, neither the NOD genetic background nor the IL2rg KO have been reported to cause any deficiency in MDSC development. Thus, the immunodeficient mouse models necessary for the establishment of the PDX models fully support the development of MDSCs. Macrophages are prominent players in the tumor microenvironment in TNBC, and unfortunately, it is not possible to investigate these cells in the PDX models described here due to the immunodeficient mouse host. Interestingly, it was recently shown that TNBCs could be subdivided according to immunosubtypes. While one subtype was enriched in neutrophils (defined using the same markers as MDSCs in this study), another exhibited high infiltration of macrophages but only a few neutrophils [10]. It might be possible that those PDX tumors with limited infiltration of MDSCs exhibited a phenotype that attracted macrophages. Interestingly, this study also showed that, upon targeting the neutrophils, an influx of monocytes (macrophage precursors) was observed. Furthermore, when immunotherapy-sensitive macrophage-enriched tumors acquired resistance to the treatment, a concomitant shift to a neutrophil-enriched phenotype was observed [10]. These studies interestingly demonstrate dynamics between these immune cell types that might be important for tumor progression and therapy responses, that unfortunately, cannot be mimicked in our PDX models. It could be interesting to investigate whether PDX tumors grown in less immunodeficient models, such as nude or SCID mice, where macrophages are functional, could provide a model to investigate macrophage recruitment and function. The TME is exceptionally heterogeneous and can comprise virtually all immune cell types, including macrophages, natural killer cells, dendritic cells, and T and B lymphocytes. While it is obvious that the absence of certain immune cell populations, including T and B cells, is a limitation of this experimental setup, it is possible to circumvent some of these limitations by administering, for instance, endogenous T cells when investigating interactions between MDSC and T cells following, e.g., checkpoint blockade treatment. Furthermore, as suggested above, other less immunodeficient strains with fully intact innate immune systems could be used to investigate the role of inflammatory responses. However, while our studies suggest that human cancer cells support the mobilization of murine MDSC, a similar cross-species interaction might not fully sustain the development of other immune or stromal compartments, such as fibroblasts, and thus the complex interplay within the TME might be different. Further studies are therefore needed to address these issues. However, when focusing on specific cell types that are present and functional, these models can be highly suitable to investigate how tumor intrinsic responses to various therapies affect the TME. For instance, using PDX models, we have recently shown that a potential unfavorable effect of DNA methyltransferase inhibitors leads to increased IL-1 expression that supports the recruitment of monocytic myeloid-derived suppressor cells to the TME [42]. Functionally, using immunodeficient nude mice with MDA-MB-231 breast cancer cells, it has been demonstrated that MDSCs expand in breast cancer bone metastases and induce bone destruction [43]. Another study examined the effect of G-CSF and MDSCs in cervical cancer grafted in BALB/c nude mice and showed that depleting MDSC using either anti-Gr-1 neutralizing antibodies or splenectomy inhibited tumor growth and enhanced radiosensitivity. Interestingly, this study also showed that ectopic expression of human G-CSF in the cancer cells increased the levels of MDSCs in the blood, spleen, and tumors, highlighting the importance of cancer cell-derived factors in MDSC mobilization in mice [44]. Thus, while various studies have demonstrated the presence and functional relevance of MDSCs in immune-deficient mice, no studies have comprehensively compared the ability of clinically different PDX models to mobilize MDSCs both in the primary tumor and in metastases.

## 5. Conclusions

We demonstrate that human breast cancer PDX tumors support recruitment and differentiation of MDSC in both primary tumors and metastases, and thus represent a versatile tool for studying and targeting MDSCs in breast cancer.

## Figures and Tables

**Figure 1 cancers-14-06153-f001:**
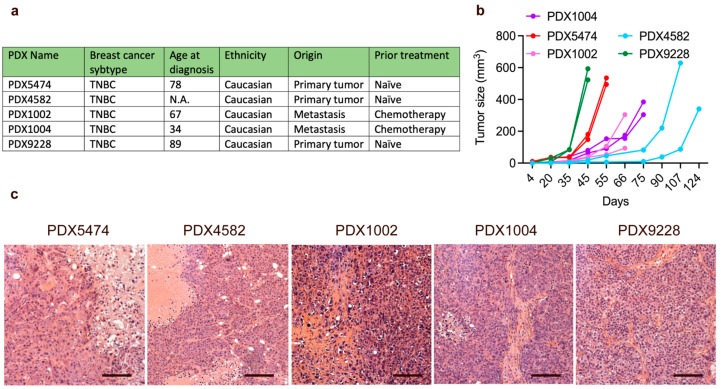
Characteristics of breast cancer PDX models. (**a**) Selected clinical parameters of the PDX models. (**b**) PDX tumor tissue was transplanted into the mammary fat pad of NOG mice, and PDX tumor growth was evaluated using caliper measurements. Each graph represents a single PDX tumor. (**c**) Hematoxylin and eosin (H and E) stainings of the breast cancer PDX models. Scalebars = 200 μM. For the H and E stainings, the PDX tumor size and timepoint upon harvest are shown in Appendix A.

**Figure 2 cancers-14-06153-f002:**
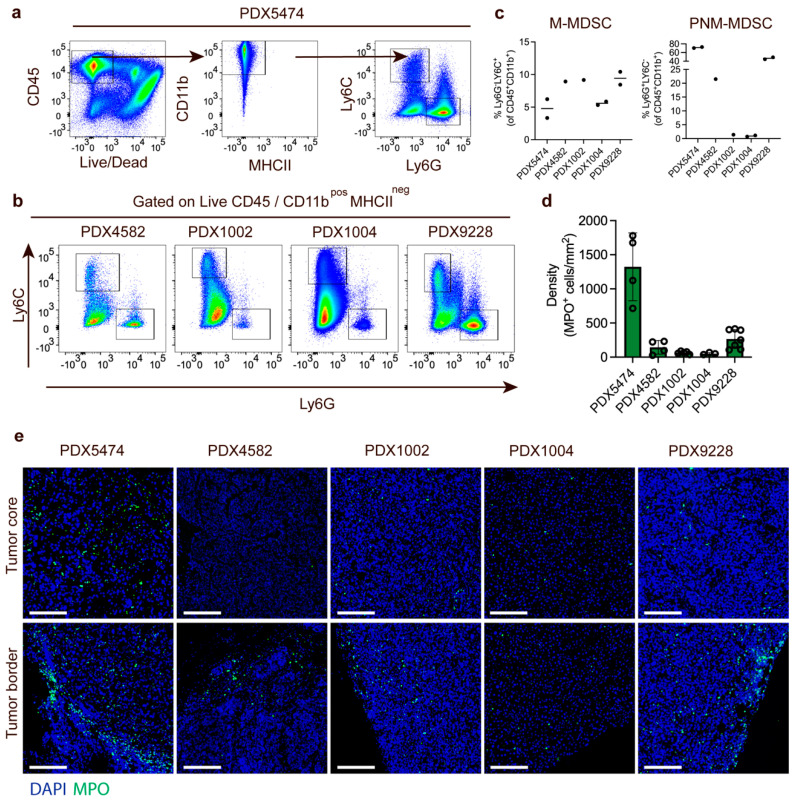
Different levels of MDSC recruitment to breast cancer PDX models. (**a**) Gating strategy for quantification of M-MDSC and PMN-MDSC in PDX tumors using PDX5474 as an example. (**b**) FACS plot showing the distribution of PMN-MDSC (Ly6C^−^LyG6^+^) and m-MDSC (Ly6C^+^LyG6^−^) in PDX4582, PDX1002, PDX1004, and PDX9228 tumors. (**c**) Quantification of M-MDSC and PMN-MDSC relative to the CD45^+^CD11b^+^ cells in PDX5474 (n = 2), PDX4582 (n = 1), PDX1002 (n = 1), PDX1004 (n = 2), and PDX9228 (n = 2) tumors. (**d**) Quantification of MPO-positive cells in immunofluorescence-stained tumor sections from PDX5474 (n = 4), PDX4582 (n = 4), PDX1002 (n = 5), PDX1004 (n = 3), and PDX9228 (n = 7). Data are presented as the number of cells per mm^2^. Four randomly selected areas from each tumor were used for the quantification. (**e**) Representative immunofluorescence staining of the PDX5474, PDX4582, PDX1002, PDX1004, and PDX9228 tumors showing myeloperoxidase-positive cells (MPO, green) and nuclear staining (DAPI, blue). For each tumor, images showing the tumor border and tumor core are presented. Scalebars: 100 μM. The PDX tumor size and timepoint upon harvest are shown in Appendix A.

**Figure 3 cancers-14-06153-f003:**
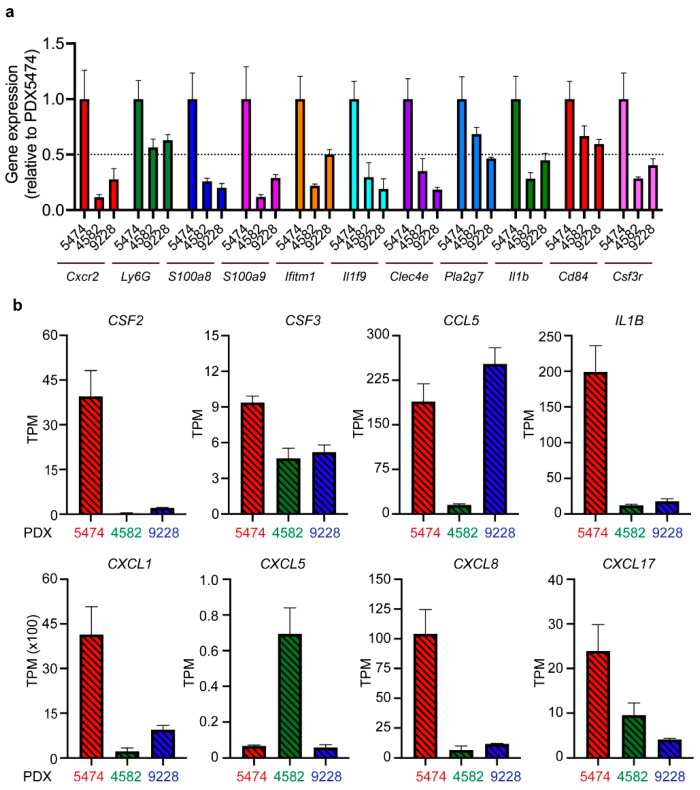
RNA sequencing analysis reveals differences in the expression of MDSC-associated genes and MDSC recruiting cytokines between different PDX tumors. (**a**) The expression levels of murine MDSC-associated genes in PDX4582, PDX5474, and PDX9228 were analyzed by RNA sequencing (n = 4). Expression data (transcriptions per million (TPM)) are presented as relative values normalized to the TPM values in PDX5474. (**b**) The expression levels of human MDCS recruiting factors in the same samples as in (**a**).

**Figure 4 cancers-14-06153-f004:**
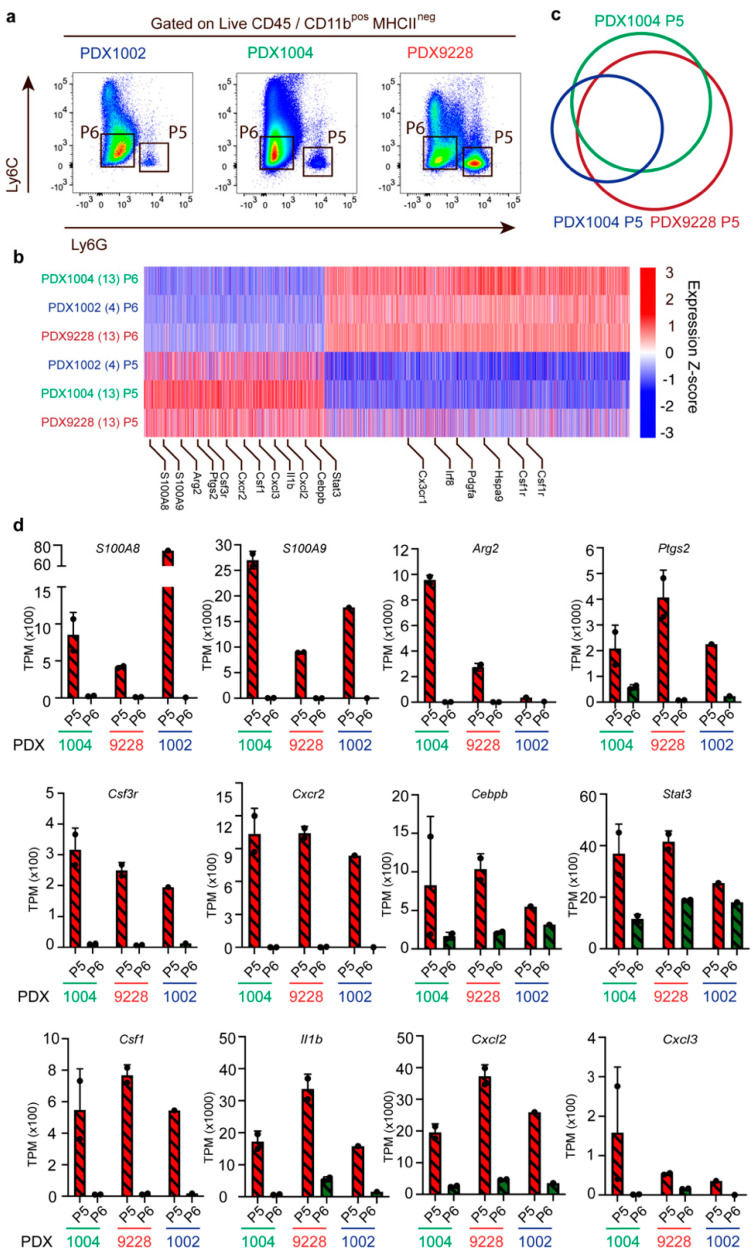
PMN-MDSC (CD11b^+^Ly6C^−^LyG6^+^) exhibit many molecular features related to immunosuppression. Transcriptomic profiling of FACS-sorted PMN-MDSCs (P5) and non-MDSC (P6) cells in the PDX1002 (n = 1), PDX1004 (n = 2), and PDX9228 (n = 2) models. (**a**) Example of the sorting gates used for the isolation of PMN-MDSC (P5) (CD11b^+^Ly6C^−^LyG6^+^) and non-MDSC (P6) (CD11b^pos^Ly6C^−^LyG6^−^). (**b**) Heatmap of 6955 differentially expressed genes between the PMN-MDSC (P5) and non-MDSC (P6) populations in PDX1002, PDX1004, and PDX9228. Genes were considered differentially regulated when the FDR < 0.05 and the log2 fold change was > 1. (**c**) Venn diagram showing the overlap of genes with increased expression in the PMN-MDSC (P5) population between PDX1002, PDX1004, and PDX9228. (**d**) Expression of genes associated with molecular features related to PMN-MDSC immunosuppression in PMN-MDSC (P5) and non-MDSC (P6) in PDX9228 (n = 2), PDX1004 (n = 2), and PDX1002 (n = 1). The data are presented as the mean TPM. Each dot represents one sample. TPM: transcription per million. The PDX tumor size and timepoint upon harvest are shown in Appendix A.

**Figure 5 cancers-14-06153-f005:**
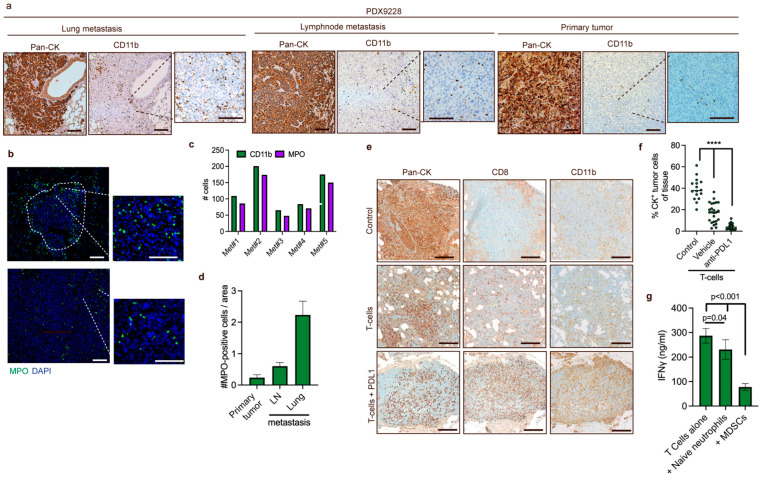
MDSCs are recruited to lung metastases, and recruitment is increased following T cell-induced inflammation (**a**) Representative IHC staining of spontaneous lung metastases in mice challenged with PDX9228 primary tumors. A pan-cytokeratin (Pan-CK) antibody was used to stain tumor cells, while a CD11b antibody was used to stain myeloid cells. (**b**) Representative immunostaining of neutrophils (green, MPO) and DNA (blue, DAPI) in the lungs from mice challenged with PDX9228 primary tumors. The tumor-stroma interface is outlined with a white dashed line. (**c**) Quantifications of CD11b-positive cells represented in (**a**) and MPO-positive cells represented in (**b**) in parallel sections of five metastases from the lungs of mice bearing PDX9228. Data are presented as the number of CD11b-positive (green bar) and MPO-positive (purple bar) cells that were counted manually in the same metastases. (**d**) MPO-positive cells were counted manually in four different randomly selected areas of primary (n = 1) and lymph node tissue (n = 1), and in four different randomly selected lung metastases (n = 1). The data are presented as the mean number of MPO-positive cells relative to the area +/− SD. (**e**) Representative images of metastasis from mice. Following establishment of metastases from tail-vein injection of MDA-MB-231 cells, mice were treated with either partly HLA-matched human T cells and anti-PDL1 antibody (T cells + PDL1) or anti-PDL1 antibody alone (control). (**f**) Quantifications of cytokeratin-positive cells represented in (**e**). Each dot represents a randomly selected lung metastasis. Data are presented as the percentage of cytokeratin positivity in each metastasis. (**g**) ELISA analysis of secreted IFNγ from T cell supernatants on 48h incubation with CD3/CD28 beads (at a cell-to-bead ratio of 1:1), naïve neutrophils from non-tumor-bearing mice (at a T cell-to-neutrophil ratio of 1:3), and MDSC from tumor-bearing mice (at a T cell-to-neutrophil ratio of 1:3). The statistical difference was determined by one-way ANOVA followed by Bonferroni’s multiple comparison testing. **** *p* < 0.0001. Scale bars: 200 μM.

## Data Availability

The NSG data generated during the study will be publicly available in the NCBI repository upon publication.

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
