# Peer review of "PDX Models: A Versatile Tool for Studying the Role of Myeloid-Derived Suppressor Cells in Breast Cancer"

_cancers, 2022, doi:10.3390/cancers14246153_

Round 1

Reviewer 1 Report (Previous Reviewer 1)

Authors have adequately addressed previous comments.

This manuscript is a resubmission of an earlier submission. The following is a list of the peer review reports and author responses from that submission.

Round 1

Reviewer 1 Report

The authors used a variety of PDX tumors to show that murine MDSCs are recruited to the tumors and that higher levels of MDSCs were found in lung metastases in these models. If appropriately characterized these models might be useful for studying the role of MDSCs in metastasis. However, a lack of rigor in the studies limits the conclusions of the manuscript.

Major concerns

Although the authors describe the rationale for using only 1-2 mice per model, more are needed to validate the model.

Discussion regarding the lack of functioning macrophages in the model should be expanded as tumor associated macrophages are key players in primary breast tumors and metastasis.

Limitations of the model should be described, including a description of the limited immune system in the model, which limits important crosstalk between immune cells.  

The time points and tumor size when tumors were harvested for H & E, flow cytometry and IF needs to be added to the figure legends and methods and/or results

Although the authors cited two references to suggest similar changes are observed in patients with breast cancer, validation in human tumors or even the TCGA database is needed. For example, what percentage of TNBC have high expression of CXCL1, IL-1B and CSF2?

Quantification of Fig 5C is not normalized based on area or total cells or per mouse and lacks error bars

The efficacy of treatment on primary tumor or number/size of mets based on Fig 5E is not convincing and is missing appropriate control groups; the sample size of the groups was not listed.

Minor concerns

The cross-reactivity and function of mouse immune cells against the human tissues should be validated.

Additional details should be provided for the table of PDXs used in Table 1 such as ethnicity and age.

A rationale is needed for why different tumors were selected for analysis in Fig 3 vs Fig 4.

Multiple grammatical and typographical errors were found throughout the manuscript.

Reviewer 2 Report

This work is highly relevant for cancer research.

Suggestion for future work - also use a control model (normal human breast tissue implanted to mice model) to differentiate between MDSC recruited to tumor by cancer characteristics of the implanted cells and MDSC recruited by non-self (human) component of the implanted cells.

At figure 2 c there is a typo error - PNM-MDSC instead of PMN-MDSC